# Recent Developments in Biological Processing Technology for Palm Oil Mill Effluent Treatment—A Review

**DOI:** 10.3390/biology11040525

**Published:** 2022-03-30

**Authors:** Debbie Dominic, Siti Baidurah

**Affiliations:** School of Industrial Technology, Universiti Sains Malaysia, Minden, Penang 11800, Malaysia; debbiedominic14@gmail.com

**Keywords:** palm oil mill effluent, bioenvironmental factor, biological treatment, fungi, bacteria, microalgae, *Lysinibacillus* sp., *Aspergillus* sp., biochemical oxygen demand, chemical oxygen demand

## Abstract

**Simple Summary:**

Palm oil mill effluent (POME) requires treatment prior to discharge to the environment. Biological processing technology is highly preferable due to its advantages of environmentally friendliness, cost effectiveness, and practicality. These methods utilized various designs and modifications of bioreactors fostering effective fermentation technology in the presence of fungi, bacteria, microalgae, and a consortium of microorganisms. This review highlights the recent biological processing technology for POME treatment as a resource utilization.

**Abstract:**

POME is the most voluminous waste generated from palm oil milling activities. The discharge of POME into the environment without any treatment processing could inflict an undesirable hazard to humans and the environment due to its high amount of toxins, organic, and inorganic materials. The treatment of POME prior to discharge into the environment is utmost required to protect the liability for human health and the environment. Biological treatments are preferable due to eco-friendly attributes that are technically and economically feasible. The goal of this review article is to highlight the current state of development in the biological processing technologies for POME treatment. These biological processing technologies are conducted in the presence of fungi, bacteria, microalgae, and a consortium of microorganisms. Numerous microbes are listed to identify the most efficient strain by monitoring the BOD, COD, working volume of the reactor, and treatment time. The most effective processing technology for POME treatment uses an upflow anaerobic sludge blanket reactor with the COD value of 99%, hydraulic retention time of 7.2 days, and a working volume of 4.7 litres. Biological processing technologies are mooted as an efficient and sustainable management practice of POME waste.

## 1. Introduction

Major cooking oils in the world are made up of palm oil, olive oil, canola oil, soybean oil, and sunflower oil. According to Oil World (2021), it is predicted that the world production of the four major vegetable oils will increase by circa seven million tonnes by 2021 and 2022 [1]. In 2021, 16,666,635 tonnes of crude palm oil were produced in Malaysia [2]. This was due to the high number of mills actively in operation within Malaysia, reaching up to 452 mills with a total processing capacity of 112.91 million tonnes of fresh fruit bunch (FFB) per year [3]. The operation of palm oil mills generates a significant amount of palm oil waste, and it is expected that 41,666,587.5 to 58,333,222.5 tonnes of POME will have been generated in 2021. This is because approximately 2.5 to 3.5 tonnes of POME are generated for every tonne of crude palm oil produced [4].

Even though Malaysia has benefited financially, palm oil milling has significantly contributed to environmental degradation both at the input and the output sides of its activities [5]. Due to the high amounts of solids, biochemical oxygen demand (BOD), chemical oxygen demand (COD), grease, and nutrients in POME, the direct discharge of the waste contributes to aquatic and land pollution. If the waste is left untreated, it can lead to rapid deoxygenation in waterbodies, thus, the ecosystem sustainability and biodiversity of the aquatic and land environments would be obstructed. The rapid growth of the palm oil industry in Malaysia and its competition with neighbouring countries such as Indonesia, has led to the tightening of environmental regulations. In this regard, the Malaysian government has taken an initiative by enacting the Environmental Quality Act 1974 to prevent, abate, and control pollution [6]. Therefore, the effluent generated from the palm oil mills needs to be treated before being discharged into the environment.

The environmental issues associated with the disposal of POME have demanded the top palm oil producer countries to reassess and re-establish their waste management policies by utilizing advanced biotreatment technologies [7]. In recent times, the treatment technologies for palm oil mill waste have been extensively re-established to ensure that palm oil mill run-off can be utilized sustainably [8]. Previously, the objective of treating the palm oil mill waste was largely for the purpose of complying with government regulations, but the awareness of needing to protect the environment has now been raised among individuals, organizations, and governments. The utilization of various industrial wastes for the production of other value-added products via biological processing is deemed practical and has been widely applied by many researchers [9,10,11,12,13,14]. This is because those wastes have the potential to be utilized due to degradable organic compounds, whereby a net positive energy gain could be achieved with a proper utilization strategy [15].

Palm oil mill waste treatment technologies have been fostering biological microorganisms, physicochemical methods, coagulation, membrane, and thermochemical processes. Biological processing for POME treatment offers low-cost, practical, and easy procedures [8], while other techniques that are implemented to treat POME include anaerobic ponding systems, integrated anaerobic–aerobic bioreactors, coagulation and flocculation, vermicomposting, membrane filtration, moving bed biofilm reactors, and zero liquid discharge [16]. The common biological processing involves anaerobic treatment, aerobic digestion, and fermentation [8], but the selection of the biological treatment methods is dependent on the ability to reduce the BOD, COD, and organic nutrients in the waste discharge.

Although there have been many promising achievements at the laboratory or pilot scale, there are several challenges to implement the POME biological treatment at the industrial scale. One of the major hindrances is the upscaling of the bioreactor, which involves the considerations of cost production, total working volume, hydraulic retention time (HRT), practicality, and processing technology effectiveness in the presence of microorganisms. By combining several strains of microorganisms with certain bioreactor designs, however, researchers have shown remarkable reduction in the BOD, COD, and organic contents prior to discharging the waste. Meanwhile, chemical and physical treatment approaches encounter limitations in terms of harmful chemical utilization and the occurrence of pore blocking at the membrane filtration surface.

The main objective of this review is to provide a summary of the recent and practical innovation of the biological processing technologies for POME treatments by highlighting the advantages and challenges within the past 15 years. Withal, the optimum treatment outputs with a low BOD, COD, and total suspended solids (TSS) will also be used as guidelines for selecting the effective biological processing technologies. In this regard, the most effective biological treatment processes concerning POME can be identified. Science Direct and Google Scholar were searched until January 2022 using a combination of search terminology such as biological, palm oil mill effluent, treatment, fungi, bacteria, microalgae, and consortium of microorganisms. Among all the inventive panoply of literature reviewed, only a final selection of 161 papers fostering the most effective in terms of removal efficiency and innovative biological processing technologies for POME as a resource utilization was identified and included as references.

## 2. Characterisation of POME

Every tonne of crude palm oil produced generates approximately 2.5 to 3.5 tonnes of POME [4]. POME is the only liquid waste produced from palm oil processing, which can be characterized as a brownish sludge with high viscosity that is composed of fine cellulosic materials, oil, and water. The brownish colour of the sludge is attributed to the fulvic acid-like components, and humic acid [17]. POME is generated from the sterilization of FFB, clarification of crude palm oil (CPO)and hydro-cyclone separation of the kernel, and can be obtained from the clarification of wastewater, sterilizer condensate, and hydro-cyclone wastewater [18,19,20]. The general characteristics of POME from different sources are tabulated in Table 1.

## 3. Biological Processing Technologies for POME Treatment

Biological processing technology implies the use of microorganisms to degrade complex organic matters present in the wastewater. Alternatively, it is also termed as a secondary treatment. The purpose of biological processing technologies for wastewater treatment is to remove pollutants such as organic carbon, nutrients, heavy metals, suspended solids, and inorganic salts by degradation biologically in the presence of microorganisms. The complex organic matter in the wastewater is oxidized into the cells of the microorganisms such as bacteria, fungi, or algae under anaerobic or aerobic conditions and subsequently eliminated by the removal process or sedimentation [23]. The sediment can be valorised to other value-added products such as biomass fuel [24].

The biological reaction happens in the bioreactor. Generally, the wastewater will be introduced into a designed bioreactor in which the organic matter will be utilized by the microorganisms. The design of the bioreaction is dependent on the required end product. This is because a high yield of the bioprocess can be achieved once an optimum external environment meets the needs of the biological reaction system [24].

Bacteria are the most typical microorganisms responsible for the stable end product of the biochemical decomposition of wastewaters [25]. Nutrients, organic substances, and pollutants available in the wastewater are utilized as food by the microorganisms to carry out metabolism. The microorganisms decompose the organic matter through two different multitudinous bioconversion routes, namely, biological oxidation and biosynthesis [26]. The biological oxidation forms some end-products, such as minerals, that remain in the solution and are discharged with the effluent (Equation (1)) [22].

Biological oxidation:(1)$$\mathrm{COHNS}\;\left(\mathrm{organic}\;\mathrm{matter}\right)+O_{2}+\mathrm{Bacteria}\;\to {\;\mathrm{CO}}_{2}+{\;\mathrm{NH}}_{3}\;\left(\mathrm{with}\;\mathrm{energy}\right)+\;\mathrm{End}\;\mathrm{products}$$

The biosynthesis transforms the colloidal and dissolved organic matter into new cells which appears in the form of dense biomass that can be removed by sedimentation (Equation (2)) [22].

Biosynthesis:(2)$$\mathrm{COHNS}\;\left(\mathrm{organic}\;\mathrm{matter}\right)+{\;O}_{2}+\;\mathrm{Bacteria}\;\to {\;C}_{5}H_{7}{\mathrm{NO}}_{7}\;\left(\mathrm{new}\;\mathrm{cells}\;\mathrm{or}\;\mathrm{biomass}\right)$$

Biological processing methods can be categorized into two methods, the anaerobic method and the aerobic method. The classification of the above-mentioned biological treatment methods is based on the content of dissolved oxygen in the wastewater. Aerobic biological treatment takes place in the presence of oxygen, while anaerobic biological treatment takes place in the absence of oxygen.

Aerobic biological treatment includes aerobic bioreactors, activated sludge, percolating or trickling filters, biological filters, rotating biological contactors, and the biological removal of nutrients [22]. The presence of oxygen allows the microorganisms to convert the organics and carbon dioxide into new biomass. In the past, aerobic biological treatment was aimed at the oxidation of organic material (collectively measured as BOD) and the oxidation of ammonium (NH_4_^+^) [27]. At present, aerobic biological treatment is commonly used to polish the industrial wastewater pre-treated by anaerobic processes [28]. The application of aerobic biological treatment after the pre-treatment of the industrial wastewater will guarantee that the wastewater is fully degraded. Concomitantly, the industrial wastewater can be discharged safely in compliance with strict environmental regulations.

Anaerobic biological treatment includes anaerobic lagoons and anaerobic bioreactors. This treatment is commonly applied for high strength wastewater with high biodegradable organic matter whereby the aerobic treatment would be inefficient. This is because the oxygen demand for the aerobic condition to be maintained during the treatment of wastewater could not be fulfilled [26]. The anaerobic treatment offers many advantages when compared to aerobic treatment, viz., a low energy input, low nutrient requirement, and the degradation of waste organic material that leads to the production of biogas energy; however, the anaerobic biological treatment of POME also has its own drawbacks, such as a long HRT.

## 4. Factors Affecting Microorganism Activity during the Biological Processing Treatment

Figure 1 delineates five bioenvironmental factors affecting microorganism activity during biological processing treatments, namely, temperature, pH, dissolved oxygen, concentration of nutrients, and toxicity. Temperature is one of the main factors that affects microorganism activity during a biological processing treatment. The regulation of ambient temperature is important because it influences the outcome of a study in terms of treatment stability and performance in biological wastewater [29]. If the temperature is regulated at the optimum level, the growth and metabolism of bacteria at an excellent level can be achieved. Consequently, a high yield of the bioprocess can be obtained as the environment provided to the microorganisms during the biological treatment is fulfilling the demand of the normal biological reaction. Consequently, the temperature will affect the enzymatic reaction of the bacteria. This is because the growth of each species is dominantly determined by the specific temperature of its enzyme. If the temperature is high, the enzymatic reaction of the bacteria will increase. Conversely, the enzymatic reaction of the microbes will reduce when exposed beyond the maximum temperature limit due to the denaturing of the enzyme. Moreover, the enzymatic reaction of the microbes will decrease if the temperature is lowered; however, some microorganisms have the ability to adapt to new living conditions rapidly after the temperature is lifted from the mesophilic range to the thermophilic range while some would just vanish from the system due to their low tolerance to high temperatures [29]. Microorganisms that possess high-temperature tolerance would be able to grow, but significantly, when there is no microbial growth, there will be no soluble COD removal either [30]. Thus, an optimum temperature needs to be regulated to ensure the effective biological treatment of wastewater.

The growth and distributions of microorganisms are influenced by pH during biological degradation. A study conducted by Nwuche et al. (2014) evinced the pH of the medium was found to affect the efficiency of COD removal from the POME [31]. The biological macromolecules activities of the microorganisms depend on the pH value [32] and bacteria have a narrower pH range for growth as compared to fungi [33]. The pH may affect the bacteria’s thermodynamic force of chemical reactions involving protons as metabolites and the energetic metabolism, provided that the proton motive force is used to be the main source of electrochemical potential for ATP synthesis [32]. The pH below the normal physiological range tolerated by the bacterial cells leads to bacterial growth inhibition [33]. Similarly, a pH above the normal physiological range tolerated by the bacterial cells will deter the bacterial growth. Moreover, increasing or decreasing the environmental pH by one unit in natural environments will lower the metabolic activity of microbial communities by up to 50% [34]. Withal, the biodegradation of POME using a consortium of microorganisms in unfavourable pH conditions will obstruct the growth of bacteria. This is because the bacteria will be outcompeted by other bacterial species that are more adapted to the pH conditions [34]. The optimum pH is upmost required to maintain the microorganisms’ metabolism, physiological mechanisms, and structural integrity during the biological degradation of the POME.

A study conducted by Tajuddin et al. (2004) [35] evinced the growth of the bacteria can be accelerated and the aerobic digestion process period can be shortened, with the increase of oxygen concentration. Liao et al. (2011) [36] also reported that a higher degradation efficiency can be achieved with a higher dissolved oxygen (DO) level. This observation pinpoints that the oxygen supplied to the microorganisms through aeration will allow them to respirate and satisfy the BOD [35]. The aeration would also increase the oxygen concentration and at the same time enhance the microorganisms’ growth, thus, reducing the COD concentration [35]. Moreover, the oxygen acts as an agent in the oxidation of undesirable contaminants [35]. As the DO concentration increases, the pollutant removals and biomass production increases [37]. If the concentration of DO is low during the biological treatment, the microbes are unable to degrade the organic matter. A sufficient concentration of DO is required by the microorganisms to degrade the copious organic matter present in the POME.

The ability of microorganisms to grow also depends on the concentration of toxicity and exposure. The toxicity for all compounds is enhanced when the acid concentration increases [38]. The high toxicity will inflict undesirable low microorganisms’ growth rates and a reduction in cell concentrations, while the severe growth inhibition of the microorganisms will disrupt the biodegradation of organic matter present in the POME. Most importantly, the target action of the toxicity is dependent on the concentration of the toxic chemical [39]. A surfeit of toxic chemicals will alter the molecular structure of the microorganisms and subsequently revise the mode of action during the biodegradation of organic matter. In lieu, at a minute concentration, the toxic chemical can be utilized as nutrients [39].

Nutrients play a significant role during microbial growth. It is extremely necessary for nutrients to be present in any media to ensure microorganisms are properly cultivated in the laboratory as they are in their natural environments. There are many types of nutrients available for proper growth and the supply of nutrients is dependent on the requirement of the study itself. The nutrients supplied will be used by the microorganisms principally in their metabolic processes and for their cellular needs. The types of nutrients essential to be supplied must contain a fermentable sugar such as carbon and energy sources for the microorganisms. The availability and concentrations of the nutrients within the microorganisms’ environment could become a factor that determines their development. In the case of an inadequate amount of nutrients supplied, the growth process will be retarded. In lieu, with sufficient availability and ideal concentrations of the nutrients, the bacterial growth will thrive. At high concentrations, the specific growth rate is independent of the concentration of nutrients, but at low concentrations, the specific growth rate is a strong function of the nutrient concentration [40]. In this regard, the nutrients could become a limiting factor during the microorganisms’ growth.

## 5. Biological Processing Technologies for POME Treatment

This subtopic highlights the effective biological processing technologies with various designs and modifications of bioreactors applicable to POME treatment, as delineated in Table 2.

### 5.1. Upflow Anaerobic Sludge Blanket Reactor (UASBR)

An upflow anaerobic sludge blanket reactor (UASBR) is a type of anaerobic digestion treatment utilized for industrial effluent treatment. Up until 2011, there were approximately 500 UASBRs installed worldwide [80]. The main highlight of the system is its integration of biological and physical processes consisting of granules. The formation of granules in POME treatment is associated with the presence of acetate in the POME itself. The formation occurs when the POME to be treated using an UASBR is concentrated with butyrate. The butyrate will be degraded by acetogenic bacteria into acetate by nature and the formation of granules can be controlled by adding phosphorus and nitrogen into the influent stream.

This system is affordable and a positive energy balance of the anaerobic treatment processes is achievable [81]. The physical process of the system involves the separation of solids and gases from the liquid simultaneously, and the biological process of the system involves the degradation of decomposable organic matter under anaerobic conditions [82]. Ahmad successfully removed 99% of COD from POME with the application of an UASBR within 7.2 days with a working volume of 4.7 L at 37 °C [41].

The UASBR has been touted as a reactor with a high loading capacity, enabling a high treatment rate of POME [82]. When the chemical and physical conditions for sludge flocculation are suitable, a high efficiency of COD removal from the POME can be achieved. Similarly, an excellent settling characteristic of the POME also contributes to the high COD removal. Withal, this study also successfully produced a higher methane content, circa 70–80%. High biogas production was also observed with an increase of organic loading rates [83] and the capability of the UASBR to support high organic loading rates during the digestion of POME increases the relevancy of the system upon application at a larger treatment scale. Additionally, the UASBR was found to be effective for the treatment of wastewater with high total suspended solids (TSS) and a maximum removal of volatile suspended solids (VSS) and TSS was obtained at a COD loading rate of 4.80 g/L/d [41]; however, the wastewater must possess a proportionately high degree of solubility for the treatment to be effective [84]. Nevertheless, it should be noted that short HRTs, high COD removal efficiency, and the application of a high organic loading capacity of the UASBR can only be achieved when the formation of granules is controlled.

### 5.2. Ultrasonic Membrane Anaerobic System (UMAS)

The ultrasonic membrane anaerobic system (UMAS) is one of the anaerobic digestion treatments applied for the biological treatment of industrial effluent. This system offers a cost-effective method for POME treatment. In this system, an ultrasonic device is applied to reduce the fouling of the membrane and to simultaneously increase the COD removal efficiency [85]. UMAS is one of the alternative methods of treating POME. Based on a study conducted by Abdurahman et al. (2013), a COD removal efficiency of 98.5% with the working volume of 200 L was achieved for 480.3 days of POME treatment by utilizing an UMAS at 30 °C [42]. The study also recorded 98.0% of COD removal efficiency within 20.3 days. There was no significant difference of the COD removal efficiency when both HRTs were compared. This was because the increased biomass concentration in the system subsequently led to the washout phase of the reactor [42]. The 98.6% reduction of COD removal was equivalent to 3000 mg/L of COD content.

The introduction of an ultrasonic device eliminates the presence of membrane fouling in the system and a high COD removal percentage can be achieved in a shorter treatment time [85]. The accumulation of particles at the membrane surface of the UMAS is reduced as the fouling effect is overcome by the presence of ultrasonic waves [85]. For that reason, the COD removal efficiency can be observed to increase gradually throughout the biodegradation of POME. In the case without the application of an ultrasonic device, fouling could build up over time while the COD removal efficiency would decrease. This is because the resistance of the POME flowing over the surface would increase, due to the reduced cross-sectional area of the flow channels. Nonetheless, the COD removal efficiency from the POME by the UMAS is lower compared to the COD removal efficiency from the POME by an UASBR. This is because the longer the HRTs, the greater the fouling effects and along with the biological treatment of POME using a UMAS, huge deposits of solids would form on the membrane surface. Concomitantly, the membrane pores will become blocked and the performance of the UMAS system will be reduced as compared to the UASBR.

### 5.3. Membrane Anaerobic System

A membrane anaerobic system (MAS) is an anaerobic treatment of wastewater in the presence of a cross flow ultra-filtration membrane. Based on the study conducted by Abdurahman et al. (2011), the removal efficiency of COD was between 96.6% and 98.4% with HRTs from 6.8 to 600.4 days [43]. Approximately 98.4% of the COD was removed during the biological treatment of POME with a working volume of 50 L within 600.4 days. The COD removal efficiency was higher compared to the 82% of COD removal observed for the POME treatment using a sequencing batch reactor (SBR) system. The COD removal efficiency using the MAS reported by Abdurahman et al. (2011) was lower compared to the COD removal efficiency using a UASBR and UMAS [41,42]. This could be due to the deposition of suspended solids and the formation of granules on the membrane of the MAS.

The total methane production from the MAS was also found to be lower compared to the COD removal efficiency by the UASBR and UMAS. This observation was due to the presence of high suspended solids contents in the POME. According to Zouari et al. (2015), the suspended solids in wastewater are responsible for the decrease of the methanogens count [86]. In the event of the suspended solids from the wastewater accumulating and subsequently the forming of a coarse non-biomass, the active biomass will be diluted. Concomitantly, the number of methanogens reduces, leading to a low methanogenic capacity during the biological treatment process of POME. Nevertheless, in the study, the VSS fraction increased to 85% [43]. This observation pinpoints that the long HRT of the MAS facilitated the decomposition of the suspended solids and their subsequent conversion to methane [43]. The application of OLR also correlates to methane content during the biological treatment of POME. High OLR in the system will lead to the decrease of methane content and this is due to the favourable proliferation of acetic acid bacteria as compared to methanogenic bacteria [43]. Nonetheless, these membrane separation techniques have been proven to be an effective method for separating biomass solids from digester suspensions and recycling them to the digester [87].

## 6. Biological Treatment of POME Using Microorganisms

Various microorganisms have been applied for the POME treatment process, such as fungi, bacteria, and microalgae. This section will elaborate in detail the application of various microorganisms for effective and practical POME treatment by taking into account the COD removal, working volume, temperature setting, and treatment times.

### 6.1. Biological Treatment of POME Using Fungi

Fungi have also been utilized in biological processing technologies for POME treatment, as delineated in Table 3. 

*Yarrowia lipolytica* is a non-pathogenic marine hydrocarbon-degrading yeast. Based on a study conducted by Oswal et al. (2002), *Y. lipolytica* NCIM 3589 isolated from Mumbai, India successfully removed 97.40% to 97.80% of COD from POME within three to four days without dilution of the POME and nutrient supplement [88]. Specifically, approximately 5500 mg/L (equivalent to 97.80% COD removal efficiency) of COD was removed efficiently from the POME within four days. There was no significant difference of the COD removal efficiency when the POME was treated with *Y. lipolytica* for three and four days. A high COD removal efficiency within the designated HRTs indicated that the yeast possessed many advantages which included a wide range pH tolerance. The POME sample used in Oswal et al.’s study was initially acidic. Notably, the yeast was capable to grow between pH 3.0 and 8.5 [88]. This was due to the yeast having developed a uniquely broad spectrum of biological features that enabled them to grow in different environments as compared to their native environment. Moreover, with a high efficiency of COD removal from the POME, *Y. lipolytica* has emerged as a paradigm organism to produce several advanced biofuels and chemicals [98].

The yeast is an oleaginous microorganism also known as the degrader of alkanes in crude oil that has the ability to accumulate lipids [99,100]. It does not only serve as a biological treatment of POME, but it also can serve as a biological tool for biofuel production. The presence of alkanes, fatty acids, grease, and triacylglycerols in POME can be utilized by *Y. lipolytica* as food during the biological treatment of POME. *Y. lipolytica* also has the ability to remove oil and grease from POME due to the production of enzymes. For example, the production of lipases and enzymes can partially or completely metabolize these compounds [101]. Generally, the enzymes are produced in the presence of oils or inducers including triacylglycerols, fatty acids, hydrolysable esters, Tweens, bile salts and glycerol [102]. The expression of lipase activity has frequently been caused by the carbon source [103], while the direct transesterification of yeast lipid by its own lipase also indicated a potentially high use of this yeast in environment-friendly biodiesel production [103]. Substantially, the degradation of the POME sample from a factory site in India recorded a 99% (1500 mg/L) COD reduction utilizing the POME sample sequentially treated with flocculant, ferric chloride and then with a consortium of microorganisms developed from a local garden.

*Trichoderma viridae* is an oil borne, green-spored ascomycetes known to be abundant in the environment [104]. It is also regarded as the most abundant colonizer of cellulosic materials, and it can frequently be found wherever decaying plant material is available [105]. Biological treatment of POME in the presence of *T. viride* carried out by Karim et al. (1989) successfully achieved more than 95% of COD removal efficiency of 0.3 L POME within 10 to 14 days at 28 ± 2 °C. Based on Karim et al.’s study, approximately a final value of 44.0 to 56.0 mg/L of COD remained in the POME [89]. The current standard discharge limit of POME as justified by Department of Environment in Malaysia is 100 mg/L of COD. In conjunction with the POME treatment study by Karim et al. (1989), the biological treatment of POME using *T. viridae* can be considered as one of the potential POME treatments [89]. This is because the remaining COD in the POME after biological treatment by *T. viridae* was compliant with the current standard discharge limit by the DOE. The decomposition of POME increases when *Trichoderma* spp. is utilized and *Trichoderma* spp. are found to increase the rate of decomposition of POME, thus, reducing the timeframe from 4–6 months to 21–45 days [106]. In this regard, *Trichoderma* spp. are touted as a good natural decomposition agent due to their ability to escalate the rate of the organic matter decomposition process.

Withal, *Trichoderma* spp. have the ability to produce enzymes that degrade the components of the cell wall. The POME contains solids composed of lignocellulosic debris from palm oil mesocarp [107]. The enzymatic activities of cellulase and hemicellulose promote the degradation of cellulose and hemicellulose that help to reduce the time of the decomposition process [108]. Siddiquee et al. (2017) has proven that *Trichoderma* spp. could increase the rate of decomposition leading to the high availability of nutrients in soil utilized by other organisms [109]. *Trichoderma* spp. can be considered as a potential natural decomposing agent that can produce a high quality of compost with an aerobic or microaerobic, low oxygen concentration condition but which is not quite anaerobic [110]. The application of *T. viridae* for the biological treatment of POME was found to not be effective from the perspective of organic load reduction because *T. viridae* is not indigenous to POME [111,112]. It was suggested that any microorganisms used for the biological treatment of POME should be indigenous and this is because the indigenous microbial isolates from the POME have been observed to possess the metabolic potential to degrade organic components [113].

*Saccharomyces* sp. is a facultative alcoholic yeast [114]. *Saccharomyces cerevisiae* has been widely cultured in several waste feedstocks including cassava and POME [115]. The yeast has the ability to eliminate the long duration for a start-up process and grow rapidly within a POME environment [58]. For example, *Saccharomyces* sp. L31 was locally isolated from dried POME and soil surrounding the POME dumpsite and palm wine in Nigeria where Iwuagwu et al. (2014) successfully achieved 83% COD removal efficiency from POME with the HRTs of four days [90]. Thus, the locally isolated yeast in this study proved its ability to reduce the COD in POME and to concomitantly proliferate in the POME. The *Saccharomyces* sp. L31 used the POME as the carbon source to achieve a waste-to-value product of feed grade yeast with a reduction in pollution potential [90]. The yeast conducted a fermentative metabolism to regenerate the NAD+ coenzyme and to make energy, subsequently producing carbon dioxide and ethanol [116]; however, the degradation of the POME by *Saccharomyces* sp. L31 was found to not be as effective as with *Yarrowia lipolytica* and *Trichoderma viridae.* This was because the *Saccharomyces* sp. L31 was not indigenous to the POME [111,112]. The volume of POME biologically treated using the *Saccharomyces* sp. L31 was the smallest compared to the volume of POME biologically treated using *Trichoderma viride* ATCC 32,086 as conducted by Karim et al. (1989) [89]. Despite the smaller volume of POME being biologically treated, however, the reduction in COD by the *Saccharomyces* sp. L31 can be considered economical. Iwuagwu et al. (2014) also suggested that even though the product of this process retained a higher COD than the regulatory requirement for final disposal, the reduction of COD by *Saccharomyces* sp. L31 would be considered more economical if the reduction in COD were the sole end of the intended process [90].

### 6.2. Biological Treatment of POME Using Bacteria

This subtopic highlights the effective biological processing technologies in the presence of bacteria applicable to POME treatment, as delineated in Table 4. Karim et al. (2019) successfully treated POME biologically using *Bacillus cereus* MF 661,883 with the highest COD removal efficiency at 79.35%, BOD removal efficiency at 72.65%, 0.2 L working volume at 35 °C, and HRTs of six days [117]. The highest COD removal efficiency was recorded for the 50% diluted POME prior to biological treatment by *B. cereus* MF 661883. The remaining COD and BOD in the treated POME were equivalent to 4859 ± 605 mg/L and 4054 ± 368 mg/L. The POME contained an enormous amount of organic compounds that were primarily composed of cellulolytic material originating from cellulose fruit debris. Cellulase-producing bacteria are needed to degrade cellulose, such as *B. cereus;* a spore-forming bacteria that has the capability to produce cellulase and perform the degradation of cellulose-based agro-industry waste. Bala et al. (2015) has also reported that cellulolytic activity in a liquid medium could be facilitated by *B. cereus* due to the ability of *B. cereus* to excrete enzymes [113]. The COD and BOD in POME reduced significantly after being biologically treated with *B. cereus* MF 661883. This observation indicated that the bacteria had degraded the cellulolytic material using cellulase by reducing it into sugar. The *B. cereus* MF 661,883 also utilized carbon from the POME for growth, therefore the abundance of organic compounds in POME that is responsible for the BOD and COD makes POME itself a suitable substrate for the cultivation of the various growth of microorganisms [93]. 

A study conducted by Soleimaninanadegani et al. (2014) evinced that bacterial growth increases with a significant COD concentration reduction in POME [121]. Under a similar POME treatment condition, *B. cereus* MF 661,883 produced a higher biomass and lipid content than *Rhodococcus opacus* and *Pseudomonas aeruginosa* (Karim et al. 2019). The higher biomass production by *B. cereus* MF 661,883 could be due to the organic degradation by bacteria. The bioconversion of the organic compounds into a simpler form for assimilation by the bacteria could also be attributed to a significant reduction in COD and BOD [122,123]. The high accumulation of lipid content by the *B. cereus* was attributed to wastewater assimilation and lipid accumulation by microorganisms is greatly influenced by the carbon–nitrogen ratio and biomass growth [124]. It should be noted that the biomass productivity of a microorganism does not correlate to high lipid content [125]. Karim et al. (2019) recorded 74.17% oil and grease removal via a biological POME treatment process in the presence of *B. cereus* MF661883 [117] and this was because the bacteria were able to excrete lipase enzyme for lypolytic activity [126]. This observation indicated the practicality of *B. cereus* MF661883 with a cultivation of 50% diluted POME and that it has the potential to be applied for the bioremediation of POME while simultaneously producing a higher lipid content and promoting a higher biomass growth. Thus, there is also potential for greater environmental resilience and renewable energy production using *B. cereus* MF661883.

Bala et al. (2015) successfully treated POME biologically using *Bacillus cereus* 103 PB with the highest BOD removal efficiency at 90.98%, COD removal efficiency at 78.60%, and HRTs of five days [113]. *Bacillus cereus* 103 PB is one of the indigenous bacteria locally isolated from POME in Malaysia. Among the isolated microorganisms from POME, the highest COD removal efficiency was recorded by *B. cereus* 103 PB at 78.60% followed by *Micrococcus luteus* 101 PB (67.19%), *Bacillus subtilis* 106 PB (64.08%), and *Stenotrophomonas maltophilia* 102 PB (61.92%). The study suggested that the locally isolated microorganism by Bala et al. (2015) can effectively remove at least 60% of the COD from POME [113]. The biological treatment was largely influenced by active microorganisms [111,112] where the bacteria that exist indigenously in POME utilize the organic substances which serve as nutrients. The nutrients are degraded by the indigenous bacteria into simpler by-products for growing purposes. Withal, *B. cereus* 103 PB has the ability to remove nitrogenous compounds from POME and according to Rout et al. (2018), *B. cereus* has successfully removed ammonium (NH_4_^+^-N) and nitrate nitrogen (NO_3_-N) from wastewater biologically with ammonium removal (66%) dominating over the biological nitrate nitrogen removal (61%) [127]. Moreover, the denitrification of POME could be facilitated by *B. cereus* either in anaerobic or aerobic conditions [117]. NH_4_^+^-N and NO_3_-N could be removed from wastewater through assimilation by *B. cereus* as intracellular nitrogen. Additionally, the NH_4_^+^-N and NO_3_-N could also be converted by *B. cereus* into N_2_ gas via a heterotrophic nitrification–aerobic denitrification process and an aerobic denitrification process [117,127]. Since POME is composed of mainly water, cellulolytic material and oil, the degradation of the cellulose and oil in POME is necessary by lipase and cellulase-producing bacteria such as *B. cereus* 103 PB, the strain, *B. cereus* 103 PB, has the ability to excrete cellulase and lipase extracellularly [113]. For example, during the microbial degradation of oil in POME by *B. cereus* 103 PB, the lipase hydrolyses the oil into volatile fatty acids and organic acids. Alternatively, the *B. cereus* 103 PB could beta oxidise the fatty acids into simpler metabolites via the fatty acid degradation pathway and acetyl-CoA that serves as the terminus for the production of citric acid. Further along, the cellulose in POME is degraded by the bacteria using cellulase via hydrolysis. The degradation of cellulose in POME is thus commenced by the bacteria with the aid of extracellular enzymes. Hence, the exploitation of indigenous microorganisms isolated locally from POME offers a very efficient tool to bioremediate and biodegrade POME before it being released into the environment due to the microorganisms’ metabolic ability.

Islam et al. (2016) successfully treated POME biologically using *Klebsiella variicola* with a COD removal efficiency at 74%, and HRTs of 12 days [128]. The COD removal efficiency at 74% was obtained upon subjecting the POME to pre-treatment with ultrasonication prior to biological treatment with *K. variicola*, whereas the highest COD removal efficiency of 48% was only achieved when the POME was left untreated prior to the biological treatment by *K. variicola.* A significant COD removal efficiency difference was observed between pre-treated POME and un-treated POME. *K. Variicola* is a facultative anaerobic and non-spore forming bacteria that can be isolated from various sources such as human samples, animals, insects, plants, and the environment [129]. The bacteria are potentially found to be applicable in wastewater treatment, and the biodegradation and bioremediation of polluted soil, either alone or in association with other organisms [130]. Withal, the bacteria have the capability to generate renewable energy by transforming chemical energy into electrical energy. The electricity is generated from microbial fuel cells (MFCs) enriched with pre-treated POME and inoculated using *K. variicola.* An average power density of 1648.70 mW/m^3^ and 1280.56 mW/m^3^ were obtained from pre-treated POME and un-treated POME using electroactive bacteria [76]. The large amount of electricity generated and COD removal efficiency from the pre-treated POME revealed that *K. variicola* has the competency to grow and survive in MFC conditions. Islam et al. (2017) successfully generated electricity from pre-treated POME inoculated with *K. variicola* isolated from city wastewater [131]. The study proved that the organic compounds in POME can be utilised by *K. variicola* as nutrients to generate electricity. The availability of the nutrients ensured the bacteria could successfully utilize the final discharge of POME without the additional supplementation of nutrients. Furthermore, the bacteria were able to heterotrophically nitrify and aerobically denitrify samples from different sludge environments [130]. Thus, the utilization of *K. variicola* in the treatment of POME could benefit the environment and the economy.

### 6.3. Biological Treatment of POME Using Microalgae

This subtopic highlights effective biological processing technologies in the presence of microalgae applicable to POME treatment, as delineated in Table 5. *Chlorella* sp. is a unicellular, freshwater microalgae. The single celled microalga has been present on earth since the pre-Cambrian period 2.5 billion years ago [132]. Nowadays, the microalgae are widely applied and consumed as food supplements and their products are used for various purposes such as animal feed, pharmaceuticals, food colourings, dyes, pharmaceuticals, aquaculture and cosmetics. The microalgae consist of a 20–30% lipid content [133]. *Chlorella* sp. was able to achieve between 95–99.9% COD removal efficiency, 97–99.9% BOD removal efficiency and 78–98% of total nitrogen removed with HRTs of 14 days [133]. The highest COD, BOD and nitrogen removal efficiency were achieved in the condition of immobilization in alginate beads prior to POME treatment with *Chlorella* sp. at 50%. The results were in good agreement with Mohammed et al. (2016) who found that the removal efficiency of COD can be increased to 70–80% with the utilization of *C. vulgaris*. The microalgae species has been proven to remove COD, phosphorus and nitrogen with various HRTs either being co-cultured with or without bacteria [134]. This could be due to good photosynthetic activity by *C. vulgaris* and an increased growth rate.

The dilution of POME also affects the COD, BOD, and nitrogen removal efficiency by *C. vulgaris*. Once the organic and nutrient concentration of POME was in the suitable range to be assimilated by *C. vulgaris*, the highest COD, BOD and nitrogen removal efficiency from POME could be achieved. The dilution of POME reduces the organic and nutrient concentrations to the acceptable concentration range by the microorganism. In the case of undiluted POME, an excessive amount of nutrients can reduce the levels of DO in wastewater. Predictably, the efficiency of removing COD, BOD, and nitrogen from POME by microalgae will be obviated if there is no aeration in the diluted wastewater during the biological process treatment. The utilization of *C. vulgaris* in carbon dioxide removal system was also successfully conducted by Burlew (1953) at a large scale [142]. The carbon dioxide abatement system using *C. vulgaris* was built next to a power plant that released an enormous amount of carbon dioxide whereby the biomass absorbed nitrogen from the atmosphere in the form of NOx [143]. Thus, the potential application of *Chlorella* sp. in the production of biodiesel is remarkable due to its ability to accumulate a high lipid content. For example, POME with 1 gr/L of urea showed a higher specific growth rate (0.066/day) [133,144]. In this regard, microalgae are considered as a promising sustainable and renewable energy resource.

Kumar et al. (2011) successfully utilized *Spirulina platensis* to achieve a 93.57% COD removal efficiency, 97.18% BOD removal efficiency and 86.98% of total suspended solids removal with HRTs of six days [135]. The POME sample was clarified using commercial polymers prior to treatment by *S. platensis* and the suspended solids in the POME were effectively reduced from 192 mg/L to 25 mg/L using the polymers, GPF8111 and GPF8112. The general function of the polymers was to remove the suspended solids from the POME by inducing flocculation. Upon the addition of the polymers into POME, the suspended solids aggregated to form precipitation and were removed prior to treatment by the microalgae. The clarification of the POME prior to the treatment by *S. platensis* is recommended to allow the penetration of light for the microalgae to grow. *S. platensis* is photosynthetic in nature and the *Spirulina* sp. requires bright sunlight to grow [145,146]. The effect of photo inhibition results in low productivity and deters the cell concentration of microalgae [146]. Thus, light is able to penetrate through clarified POME and facilitate the growth of *S. platensis*. The *S. platensis* is found to efficiently reduce the BOD to qualify for the discharge standards and tolerate the chemical parameters of POME. The treatment of POME using *S. platensis* also reduced the COD from 2830 mg/L to 182 mg/L and the BOD from 1490 mg/L to 42 mg/L. The heavy metal content was found to be reduced to notable levels and this observation highlights the biosorption properties at a high capacity and rapid rate [135]. The ability of *S. platensis* to accumulate hazardous compounds in the POME sample, such as pollutants, heavy metals, and ions in their cells, increases the potential of the microalgae species to be utilized in biological treatment owing to its ability to reduce pollution attributed by the release of POME into the environment. In other research, the improved production of lipid contents (182 mg/L) by cultivating *Chlorella pyrenoidosa* in POME was achieved with a COD value of 700 mg/L [147].

Hadiyanto et al. (2013) reported that *Chlorella sorokiniana* has the ability to convert COD into carbon sources to be utilized for photosynthesis [148]. Haruna et al. (2017) successfully utilized *C. sorokiniana* to achieve a 90% of COD removal efficiency, and 71% total nitrogen removal with HRTs of 15 days [136]. The highest COD removal efficiency and high total nitrogen removal indicated that the microalgae actively degraded the 80% diluted POME. The COD removal efficiency in 20, 40 and 60% dilutions of POME were 19, 27 and 86%, respectively. The COD removal efficiency by *C. sorokiniana* was found to be efficient in diluted POME compared to the concentrated POME. Chen et al. (2020) suggested that pre-dilution of wastewater is required prior to treatment in most studies [149] and a lower COD removal efficiency was observed in low diluted POME due to less volatile organic carbon removal during aeration [150]. Conversely, a higher COD removal efficiency was recorded in a high diluted POME because the penetration of light allowed the microalgae to adopt a phototrophic growth mode instead of a mixotrophic growth mode. Therefore, a high concentration of wastewater would significantly hamper the microalgae leading to an inefficient nutrient removal. Since the nutrient concentration was high at a 20, 40, and 60% dilution of POME, the excess nutrients would have caused toxicity and obstructed the growth of *C. sorokiniana.* Thus, it can be concluded that *C. sorokiniana* is suitable to be utilized for biological processing in POME treatment with an 80% dilution.

### 6.4. Biological Treatment of POME Using a Consortium of Microorganisms

Table 6 highlights the biological processing technologies in the presence of a consortium of microorganisms for POME treatment. Mishra et al. (2016) successfully utilized two types of bacteria, namely, *Clostridium butyricum* LS2 and *Rhodopseudomonas palustris*, for hydrogen production and subsequently for POME treatment [119]. A total of 93% COD removal efficiency was achieved in two-staged sequential dark and photo fermentations with HRTs of four days. The *C. butyricum* was utilized in the first stage of dark fermentation in which the COD removal efficiency was achieved at 57%. Then, the *R. palustris* was utilized in the second stage of photo fermentation for the production of hydrogen. The utilization of organic compounds in the POME sample by *C. butyricum* LS2 promoted the production of hydrogen and growth of the bacteria. During the dark fermentation, the low production of hydrogen was attributed to a low metabolic activity by *C. butyricum* LS2 during the initial phase. This could be due to a low number of bacteria growing which also indicates a low metabolic activity. Despite this, the hydrogen content was raised to 53% when the dark-hydrogen production was completed which indicates high metabolic activity of the dark-hydrogen producer. The low COD removal efficiency recorded could be attributed to the presence of a complex organic compound in the concentrated POME. The POME was diluted and further treated with *R. palustris* and this is because the growth of the bacteria used in the treatment of concentrated POME during the first stage of fermentation could have been inhibited due to the presence of water-soluble antioxidants, phenolic acids, and flavonoids in a high amount [151]. The highest COD removal efficiency became achievable because the diluted POME allowed better light penetration for bacterial growth, especially for the *R. palustris*.

The consortium of fungi, namely, *Emericella nidulans, Aspergillus niger* and *Aspergillus fumigatus* in the biological process of the POME treatment was efficiently achieved with a 91.43% COD removal and 94.34% BOD removal with HRTs of five days [93]. The application of mixed cultures of microorganisms in the POME treatment evinced a remarkable BOD and COD reduction rate compared to the single microorganism culture in the similar study. This observation was due to the synergism between the *E. nidulans, A. niger* and *A. fumigatus.* The synergistic effects shown by the fungi is attributed to the function or ability of each fungus which advantageously complemented one another. Bala et al. (2018) also reported that an effective performance of biodegradation could be achieved due to the synergistic effect of a mixed microbial consortium for POME treatment [152]. The utilization of the mixed culture combination also displayed metabolic versatility and superiority to pure cultures [153]. In comparison to a single-species culture, the simulation of natural processes in the POME environment was only possible when there were consortium activities [154]. The excellent COD and BOD removal efficiency achieved significantly prove that the synergism of the fungi promoted their biodegradation, bioaccumulation, and bio adsorption.

**Table 6 biology-11-00525-t006:** Biological processing technologies for POME treatment using a consortium of microorganisms.

No.	Microorganism	Removal Efficiency								Parameter			Remarks	References
		COD % (mg/L)	BOD % (mg/L)	TSS % (mg/L)	OLR % (kg COD/m^3^ day)	Total Nitrogen	NH_3_-N %	Methane (CH₄) Gas Release (%)	Oil and Grease (mg/L)	°C	Working Volume (L)	HRT (day)		
1.	*Clostridium butyricum* LS2 (1st) and *Rhodopseudomonas palustris* (2nd)	93	ND	ND	ND	ND	ND	ND	ND	37 ± 1	1	4	1st stage: dark fermentation 2nd stage: photo fermentation Hydrogen production: 3.064 mL H_2_/mL	[119]
2.	*E. nidulans + A. niger + A. fumigatus*	91.43	94.34	ND	ND	ND	ND	ND	ND	30	0.25	5	POME treatment	[93]
3.	Consortium of bacteria and fungi: *Micrococcus luteus* 101 PB, *Stenotrophomonas maltophilia* 102 PB, *Bacillus cereus* 103 PB, *Providencia vermicola* 104 PB, *Klebsiella pneumonia* 105 PB, *Bacillus subtilis* 106 PB, *Aspergillus fumigatus* 107 PF, *Aspergillus nomius* 108 PF, *Aspergillus niger* 109 PF, and *Meyerozyma* *guilliermondii* 110 PF	91.06	90.23	92.23	ND	ND	ND	ND	ND	ND	1	50	POME treatment	[152]
4.	*Bacillus cereus* 103 PB and *Bacillus subtilis* 106 PB	90.64	93.11	ND	ND	ND	ND	ND	ND	37	0.25	5	POME treatment	[113]
5.	Consortium of *B. subtilis* and *A. niger*	90.3 (1000 ± 100)	ND	ND	ND	1920 ± 75	780 ± 20	ND	ND	40	0.095	7	POME treatment	[31]
6.	*Bacillus toyonensis* strain BCT-71120 and *Stenotrophomonas rhizophila* strain ep10	86	94	80	ND	ND	ND	41.05	ND	ND	3	18	Production of methane using anaerobic consortium bacteria	[155]
7.	Co-culture of yeast (*Lipomyces starkeyi*) and bacteria (*Bacillus cereus*)	83.66 ± 1.9	77.34	71.43	ND	65.30	76.59	ND	79.23	30	0.2	6	Microbial lipid accumulation: 2.27 g/L	[95]
8.	Consortium of *Scenedesmus* sp. UKM9 and *Chlorella* sp. UKM2	71.00	ND	ND	ND	ND	ND	ND	ND	25 ± 2	2	25	Integrated 2 stage treatment of POME treatment	[156]
9.	*Klebsiella variicola* and *Pseudomonas aeruginosa*	69.28	ND	ND	ND	ND	ND	ND	ND	ND	0.02	11	Electricity generation using MFC: 12.21 W/m^3^	[157]
10.	*Pseudomonas* sp. on *Chlorella sorokiniana* CY-1	53.70	ND	ND	ND	55.6	ND	ND	ND	25	ND	5	Lipid production	[123]
11.	Consortium of microalgae: *Chlorella sorokiniana* UKM2, *Coelastrella* sp. UKM4 and *Chlorella pyrenoidosa* UKM7	27.55 (2845 ± 159)	20.59 (725 ± 66)	ND	ND	22.27 (506 ± 82)	−4.10 (279 ± 14)	ND	ND	25 ± 1	1.8	7	Anaerobic pond in Dominion Square Palm Oil Mill, Gambang, Pahang, Malaysia (APDI)	[158]
12.	Consortium of microalgae: *Chlorella sorokiniana* UKM2, *Coelastrella* sp. UKM4 and *Chlorella pyrenoidosa* UKM7	25.97 (3352 ± 193)	22.65 (731 ± 52)	ND	ND	25.09 (241 ± 17)	5.58 (254 ± 33)	ND	ND	25 ± 1	1.8	7	Anaerobic pond at the Sime Darby Palm Oil Mill, Carey Island, Selangor, Malaysia (APCI)	[158]
13.	Consortium of microalgae: *Chlorella sorokiniana* UKM2, *Coelastrella* sp. UKM4 and *Chlorella pyrenoidosa* UKM7	15.93 (1953 ± 131)	13.03 (661 ± 41)	ND	ND	13.43 (464 ± 25)	−5.70 (241 ± 17)	ND	ND	25 ± 1	1.8	7	Facultative pond in Sime Darby Palm Oil Mill, Port Dickson, Negeri Sembilan, Malaysia (FPPD)	[158]
14.	Natural microflora anaerobic POME sludge	13	ND	ND	ND	ND	ND	ND	ND	30 ± 2	0.1	6	Bioelectricity generation: 85.12 mW/m^2^	[159]

°C = degree Celsius; HRT = hydraulic retention time; COD = chemical oxygen demand; BOD = biochemical oxygen demand; TSS = total suspended solids; OLR = organic loading rate; L = litre; SVI = sludge volume index; NH_3_-N = ammoniacal nitrogen; ND = no data.

Bala et al. (2018) successfully utilized a consortium of fungi and bacteria in biological processing for POME treatment and achieved a 91.06% COD removal efficiency, 90.23% BOD removal efficiency and 92.23% of TSS removal efficiency with HRTs of 50 days [152]. The microbial strains (*Micrococcus luteus* 101 PB, *Stenotrophomonas maltophilia* 102 PB, *Bacillus cereus* 103 PB, *Providencia vermicola* 104 PB, *Klebsiella pneumonia* 105 PB, *Bacillus subtilis* 106 PB, *Aspergillus fumigatus* 107 PF, *Aspergillus nomius* 108 PF, *Aspergillus niger* 109 PF, and *Meyerozyma guilliermondii* 110 PF) were isolated locally from the POME sample itself. The microbial consortium of various microorganisms was reported to be suitable for the degradation of industrial wastewaters [160]. The treatment was found to be efficient due to the microorganisms being indigenous and able to perform synergistic activities. In this regard, the bacteria and fungi would be able to maintain the natural environment of the POME and to maximize their metabolic abilities [161].

## 7. Conclusions

This review clearly depicts that the most prevalent biological treatment technique is the UASBR with a COD removal efficiency of 99%. The reported COD removal efficiency was achieved at the shortest HRT of 7.2 days in comparison with the UMAS and MAS. This is because of the integration of the physical and biological processes in the system that helps to facilitate the degradation of POME more efficiently. Withal, the utilization of UASBR in POME treatment is affordable and convenient. The highest COD removal efficiency of 97.80% was recorded by *Y. lipolytica* NCIM 3598 with HRTs of four days. Even though bacteria are commonly used in POME treatment, *B. cereus* MF 661,883 showed a lower COD removal efficiency as compared to fungi such as *Y. lipolytica* NCIM 3598. This is due to the robustness of the fungi itself and the ability of the fungi to perform anaerobic digestion of POME. Microalgae showed a remarkable performance in the POME treatment, however, the microalgae prolonged the HRTs up until 15 days as compared to the fungus, bacteria, and consortium of microorganisms, even though a COD removal efficiency of 99.9% was achieved. This is because the removal of the COD increases with a prolonged treatment time. Thus, the utilization of microalgae is considered less effective until the drawback of a long HRT can be addressed. The degradation of POME by an existing consortium of microorganisms is also applicable due to the dynamic metabolic and versatility effects among them. This biological treatment technique is in parallel with the United Nations Sustainable Development Goal number 12: to ensure sustainable consumption and production patterns. In lieu of chemical and physical treatment techniques, the biological treatment technique offers various advantages such as environmentally-friendly properties and having no harmful chemical usage.

## Figures and Tables

**Figure 1 biology-11-00525-f001:**
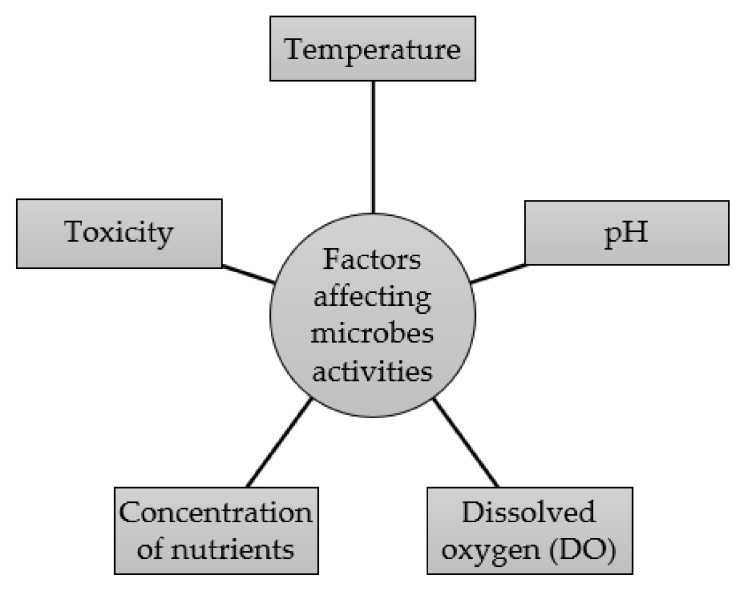
Bioenvironmental factors affecting microorganisms’ activities during biological processing treatment.

**Table 1 biology-11-00525-t001:** General characteristics of POME and its respective standard discharge limit by the Malaysian Department of Environment.

Parameters	Concentration Range		
	[20]	[21]	[22]
Chemical oxygen demand (COD)	15,000–100,000	51,000	100
Biochemical oxygen demand (BOD)	10,250–43,750	25,000	50
Total suspended solids (TSS)	5000–54,000	18,000	400
Ammoniacal nitrogen	4–80	35	100
Oil and grease	130–18,000	6000	50
Total nitrogen	180–1400	750	200
pH	3.4–5.2	4.2	5.0

All values, except pH and temperature, were expressed in mg/L.

**Table 2 biology-11-00525-t002:** Biological processing technologies for POME treatment.

No.	Biological Technique	Removal Efficiency or Concentration								Parameter			Remarks	References
		COD % (mg/L)	BOD % (mg/L)	TSS % (mg/L)	OLR % (kg COD/m^3^ day)	Total Nitrogen (mg/L)	NH_3_-N %	Methane (CH₄) Gas Release (%)	Oil and Grease (mg/L)	°C	Working Volume (L)	HRT (day)		
1.	Upflow anaerobic sludge blanket reactor (UASBR)	99	ND	ND	ND	ND	ND	70–80	ND	37	4.7	7.2	Biogas production: 20.17 11^−1^ d^−1^	[41]
2.	Ultrasonic membrane anaerobic system (UMAS)	98.5	ND	ND	0.5	ND	ND	79	ND	30	200	480.3	POME treatment	[42]
3.	Membrane anaerobic system (MAS)	98.4	ND	ND	1	ND	ND	72	ND	ND	50	600.4	POME treatment	[43]
4.	Aerobic oxidation (activated sludge reactor)	98	93	ND	ND	58	ND	ND	24	ND	91	60	Treatment of anaerobically digested POME	[44]
5.	Hybrid upflow anaerobic aludge bed (HUASB) reactor	98 (663)	ND	1387	5.5	75	23.4	ND	ND	24 ± 1	ND	47	POME treatment	[45]
6.	Anaerobic pond	97.8 (1204 ± 292)	ND	ND	1.4	ND	ND	54.4	ND	ND	ND	40	POME treatment	[46]
7.	Upflow anaerobic sludge fixed-film (UASFF)	97	ND	ND	ND	ND	ND	74.2–80.1	ND	38	4.38	3	POME treatment	[47]
8.	An integrated system of two-stage microbial fuel cells (MFCs) and immobilized biological aerated filters (I-BAFs)	96.5	ND	ND	ND	ND	93.6	ND	ND	35 ± 1	2.36	48	Direct electricity generation (input value)	[48]
9.	Upflow anaerobic sludge blanket (UASB)	96.3	ND	ND	ND	ND	ND	ND	ND	28.0 ± 2.0	10.0	20.0	Anaerobic POME treatment for methane production: 0.012 L CH_4_/g COD degraded	[49]
10.	Aerobic submerged membrane bioreactor (ASMBR)	96–98	ND	ND	ND	ND	ND	ND	ND	ND	20	8	Improved with the addition of bio-fouling reducers	[50]
11.	Combined high-rate anaerobic reactors	95.6	ND	ND	13	ND	ND	59.5–78.2	ND	36 ± 1	2	2.4	POME treatment	[51]
12.	Anaerobic expanded granular sludge bed (EGSB) reactor	94.89	ND	ND	ND	ND	ND	ND	65–70	ND	ND	9.8	Inoculum from open anaerobic ponds of POME	[52]
13.	Upflow anaerobic sludge fixed-film (UASFF) bioreactor	94	ND	ND	ND	ND	ND	(0.331)	94	50	3.65	1.5	POME treatment	[53]
14.	Anaerobic bioreactor	93.7 (2523 ± 19)	800 ± 16 (2.0)	37.9	ND	327 ± 11(3.4)	220 ± 8	ND	ND	35 ± 3	ND	100	Biogas generation: 474.6 ± 97.4 m^3^ day^−1^	[54]
15.	Lab scale sequencing batch reactor (activated sludge)	93.2 ± 1.2 (906 ± 140)	95.5 ± 1 (62 ± 28)	97.2 ± 1.3 (363 ± 190)	1.8–4.2	ND	ND	ND	ND	28 ± 1	1.8	15	POME treatment	[55]
16.	Anaerobic expanded granular sludge bed (EGSB) bioreactor	93 (1959)	ND	26,704	ND	560	64.4	43	3856	35	12	3	POME treatment	[56]
17.	Upflow anaerobic filter (UFAF) reactor	91.6	ND	ND	ND	ND	ND	ND	ND	28.0 ± 2.0	5.0	13.5	Anaerobic POME treatment for methane production: 0.482 L CH_4_/g COD degraded	[49]
18.	Anaerobic expanded granular sludge bed (EGSB) reactor	91	ND	ND	17.5	ND	ND	70	ND	35	20.5	2	POME treatment	[56]
19.	Upflow anaerobic sludge blanket-hollow centered packed bed (UASB-HCPB) reactor	90	90	80	27.65	ND	ND	60	ND	55	5	2	POME treatment	[57]
20.	Aerobic oxidation (activated sludge reactor)	89	82	ND	ND	3.0	ND	ND	112	ND	91	60	POME treatment	[44]
21.	Rotating biological contactors (RBC)	88	ND	ND	ND	80	ND	ND	ND	ND	61	5	Innoculated with *S. cerevisiae*	[58]
22.	Lab-scale sequencing batch reactor	86	87	89	ND	ND	ND	ND	ND	50	1.8	2.5	Thermophilic aerobic treatment system of anaerobically digested POME	[59]
23.	Anaerobic fluidised bed reactor	85.00	91.00	89.00	4.0	ND	ND	ND	ND	ND	2	17	POME treatment	[60]
24.	Integrated baffled reactor	83 (7735.0 ± 227.5)	ND	24,400	7.64	ND	ND	75–54	ND	32 ± 2	ND	6	POME treatment by inoculation with anaerobic pond sludge	[61]
25.	Hybrid upflow anaerobic sludge blanket (HUASB) reactor	82	ND	80	ND	87	ND	ND	ND	37 ± 1	7.22	40	POME treatment	[62]
26.	Carrier anaerobic baffled reactor (CABR)	82	ND	ND	11.38	ND	ND	75–54	ND	ND	ND	26	POME treatment by inoculation with anaerobic pond sludge and biogas production	[63]
27.	Biological sequencing batch reactor	82	ND	62	ND	ND	ND	ND	ND	ND	50	28–36	POME treatment	[64]
28.	Continuous stirred tank reactor (CSTR)	80	ND	ND	3.33	ND	ND	62.5	ND	ND	ND	18	POME treatment	[65]
29.	Continuous stirred-tank reactor (CSTR)	77	ND	ND	ND	ND	ND	ND	ND	55	1	8	POME treatment by thermophilic anaerobic reaction Methane emission: 6.05–9.82 L/day	[66]
30.	Anaerobic contact filter	73	ND	ND	ND	ND	ND	ND	ND	ND	ND	7	Biohydrogen generation: 56 L	[67]
31.	Anaerobic digestion using continuous stirred tank reactors	71.10	ND	ND	ND	ND	ND	71.04	ND	37	1.6	7	Biogas production	[68]
32.	Aerobic bioreactor	71.1 (681 ± 11)	25 ± 9 (36.0)	ND	ND	14 ± 1 (7.1)	ND	ND	ND	35 ± 3	ND	100	POME treatment	[54]
33.	MFC	70 (964)	ND	ND	ND	ND	ND	ND	ND	25–28	0.45	15	Treatment with (polacrylonitrile carbon felt) and bioelectricity generation: 22 mW/m^2^	[69]
34.	Upflow anaerobic sludge blanket fixed-film (UASB-FF) bioreactor	68	ND	ND	ND	ND	ND	ND	ND	200	2.55	1.5	Hydrogen gas: 0.31 L H^2^/g COD	[70]
35.	Upflow anaerobic filter (UFAF) reactor	66.3	ND	ND	ND	ND	ND	ND	ND	28.0 ± 2.0	5.0	1.50	POME treatment for methane production: 0.107 l CH_4_/g COD degraded	[49]
36.	Continuous stirred-tank reactor (CSTR)	66.09	ND	ND	ND	ND	ND	48.05	ND	35	4.5	12	Anaerobic methanogenic degradation of POME	[71]
37.	Upflow anaerobic sludge blanket (UASB)	65	ND	ND	ND	ND	ND	58	ND	55	1.2	5	POME treatment	[72]
38.	Upflow anaerobic sludge blanket (UASB)	62.5	ND	ND	ND	ND	ND	ND	ND	28.0 ± 2.0	10.0	2.86	Anaerobic POME treatment for methane production: 0.013 L CH_4_/g COD degraded	[49]
39.	Upflow anaerobic sludge blanket (UASB)	62	ND	ND	5.0	ND	ND	ND	ND	37	5	12	Continuous hydrogen production: 0.35 L H_2_/g COD removed	[73]
40.	Upflow anaerobic sludge blanket (UASB)	62	ND	ND	ND	ND	ND	ND	ND	37	5	0.33	POME treatment using *Clostridium* LS2 for enhanced hydrogen production: 67%	[74]
41.	Anaerobic sequencing batch reactor	62.2 ± 2.8 (26,500)	ND	93.6 ± 1.1	ND	ND	ND	ND	ND	60 ± 1	2	4	POME treatment for hydrogen production: 6.1 ± 0.03 LH_2_POME/d	[75]
42.	Membrane bioreactor	53.4 (486 ± 5)	18 ± 5 (27.8)	93.4	ND	28 ± 1 (3.6)	ND	ND	ND	35 ± 3	ND	100	POME treatment	[54]
43.	Expanded granular sludge bed reactor	53	ND	ND	ND	ND	ND	59	ND	55	1.0	5	POME treatment	[72]
44.	Microbial fuel cell (MFC)	48.63 (31,980)	46.54 (14,080)	75.27 (2882)	ND	ND	57.69 (11)	ND	ND	ND	0.02	10	Bioelectricity generation: 207.28 mW/m^3^	[76]
45.	Microbial fuel cell (MFC)	45.21 (33,200)	45 (13,200)	70.91 (2920)	ND	ND	56.52 (10)	ND	ND	25–28	0.45	15	Bioelectricity generation: 45 mW/m^2^	[69]
46.	Anaerobic ponding system	41.2	77.8	ND	ND	ND	ND	ND	ND	ND	ND	18	Zeolite performance for POME treatment	[77]
47.	Anaerobic sequencing batch reactor (ASBR)	37	ND	ND	ND	ND	ND	ND	ND	37	2	4	POME treatment	[78]
48.	Aerobic inner-circulation biofilm reactor	22 (1439)	ND	22,579	ND	238	0	ND	258	25	5	10	POME treatment	[79]

°C = degree Celsius; HRT = hydraulic retention time; COD = chemical oxygen demand; BOD = biochemical oxygen demand; TSS = total suspended solids; OLR = organic loading rate; L =litre; SVI = sludge volume index; NH_3_-N = ammoniacal nitrogen; ND = no data.

**Table 3 biology-11-00525-t003:** Biological processing technologies for POME treatment in the presence of fungi.

No.	Fungi	Removal Efficiency								Parameter			Remarks	References
		COD % (mg/L)	BOD % (mg/L)	TSS % (mg/L)	OLR % (kg COD/m^3^ day)	Total Nitrogen	NH_3_-N %	Methane (CH₄) Gas Release (%)	Oil and Grease (mg/L)	°C	Working Volume (L)	HRT (day)		
1.	*Yarrowia lipolytica* NCIM 3589	97.80	ND	ND	ND	ND	ND	ND	ND	30	ND	4	POME treatment	[88]
2.	*Yarrowia lipolytica* NCIM 3589	97.40	ND	ND	ND	ND	ND	ND	ND	30	ND	2	POME treatment	[88]
3.	*Trichoderma viride* ATCC 32086	95.00 (44.0–56.0)	ND	ND	ND	ND	ND	ND	ND	28 ± 2	0.3	10–14	POME treatment	[89]
4.	*Saccharomyces* sp. L31	83	ND	ND	ND	ND	ND	ND	ND	28 ± 2	0.025	4	Production of value-added feed grade yeast biomass	[90]
5.	*Aspergillus niger* A 103	82.1	ND	ND	ND	ND	ND	ND	ND	32	0.1	7	Production of citric acid: 5.2 g/L	[91]
6.	*Candida rugosa*	80.7	71.8	67.6	ND	ND	ND	ND	85.2	30	0.1	7	POME treatment supplemented with soybean	[92]
7.	*Emericella nidulans* NFCCI 3643	80.28	88.23	ND	ND	ND	ND	ND	87.34	30	0.25	5	POME treatment	[93]
8.	Pichia sp. SP5	73	ND	ND	ND	ND	ND	ND	ND	28 ± 2	0.025	3	Production of value-added yeast biomass	[90]
9.	*Rhizopus oryzae* ST 29	72.5	ND	ND	ND	ND	ND	ND	98.6	45	5	4	POME treatment	[94]
10.	*Lipomyces starkeyi* ATCC 56304	69.01 ± 2.3	ND	ND	ND	ND	ND	ND	ND	30	0.2	6	Microbial lipid accumulation	[95]
11.	*Rhodotorula glutinis* TISTR 5159	66.85 ± 1.57	ND	ND	ND	ND	ND	ND	ND	30	1	14	Supplemented with Tween 20 surfactant for production of lipids (38.15%) and carotenoids (125.94 mg/L).	[96]
12.	*Aspergillus niger* ATCC 9642	63.00 (6260 ± 40)	ND	ND	ND	ND	ND	ND	ND	30	1	7	Production of citric acid: 0.78 ± 0.02 g/L	[97]
13.	*Aspergillus niger*	52 (4055)	ND	ND	ND	ND	ND	ND	ND	40	0.095	7	POME treatment	[31]
14.	*Geotricum candidium*	49.1	79.8	91.8	ND	ND	ND	ND	83.6	30	0.1	7	POME treatment supplemented with soybean	[92]

°C = degree Celsius; HRT = hydraulic retention time; COD = chemical oxygen demand; BOD = biochemical oxygen demand; TSS = total suspended solids; OLR = organic loading rate; L =litre; SVI = sludge volume index; NH_3_-N = ammoniacal nitrogen; ND = no data.

**Table 4 biology-11-00525-t004:** Biological processing technologies for POME treatment in the presence of bacteria.

No.	Bacteria	Removal Efficiency								Parameter			Remarks	References
		COD % (mg/L)	BOD % (mg/L)	TSS % (mg/L)	OLR % (kg COD/m^3^ day)	Total Nitrogen	NH_3_-N %	Methane (CH₄) Gas Release (%)	Oil and Grease (mg/L)	°C	Working Volume (L)	HRT (day)		
1.	*Bacillus cereus* MF661883	79.35 (4859 ± 605)	72.65 (4054 ± 368)	65.91 (5101 ± 327)	ND	41.76 (191 ± 36)	36.92 (41 ± 11)	ND	74.17 (910 ± 458)	35	0.2	6	POME treatment at 50% dilution	[117]
2.	*Bacillus cereus* 103 PB	78.60	90.98	ND	ND	ND	ND	ND	ND	37	0.25	5	POME treatment	[113]
3.	*Klebsiella variicola*	74	ND	ND	ND	ND	ND	ND	ND	ND	0.02	12	Electricity generation from pretreated POME using MFC: 1648.70 mW/m^3^.	[76]
4.	*Bacillus cereus*	74.35 ± 1.7	ND	ND	ND	ND	ND	ND	ND	30	0.2	6	Microbial lipid accumulation	[95]
5.	*Klebsiella oxytoca*	73.40	47.51	65.59	ND	ND	64.28	ND	ND	ND	0.02	10	Continuous feeding of POME. Electricity generation using MFC: 207.28 mW/m^3^	[76]
6.	*Micrococcus luteus* 101 PB	67.19	ND	ND	ND	ND	ND	ND	ND	37	0.25	5	POME treatment	[113]
7.	*Bacillus subtilis* 106 PB	64.08	90.98	ND	ND	ND	ND	ND	ND	37	0.25	5	POME treatment	[113]
8.	*Clostridium* sp. LS2	62	ND	ND	ND	ND	ND	ND	ND	37	5	0.33	POME treatment and hydrogen production using UASB	[74]
9.	*Stenotrophomonas maltophilia* 102 PB	61.92	ND	ND	ND	ND	ND	ND	ND	37	0.25	5	POME treatment	[113]
10.	*Lysinibacillus* sp. LC 556247	50.83	71.73	42.99	ND	12.80 ± 0.08	ND	ND	12.03 ± 0.02	35 ± 2	0.3	5	POME treatment	[23]
11.	*Bacillus anthracis* strain PUNAJAN 1	47.44	39.00	ND	ND	ND	ND	ND	27	35 ± 1	1	2	Hydrogen production: 236 mL g COD	[118]
12.	*Clostridium butyricum* LS2	42	39	ND	ND	ND	ND	ND	ND	37 ± 1	1	4	Hydrogen production: 0.784 mL /mL	[119]
13.	*Bacillus cerius* 103 PB	ND	ND	71.63	ND	ND	ND	ND	85.14	37	0.25	5	POME treatment	[120]

°C = degree Celsius; HRT = hydraulic retention time; COD = chemical oxygen demand; BOD = biochemical oxygen demand; TSS = total suspended solids; OLR = organic loading rate; L = litre; SVI = sludge volume index; NH_3_-N = ammoniacal nitrogen; ND = no data.

**Table 5 biology-11-00525-t005:** Biological processing technologies for POME treatment in the presence of microalgae.

No.	Microalgae	Removal efficiency								Parameter			Remarks	References
		COD % (mg/L)	BOD % (mg/L)	TSS % (mg/L)	OLR % (kg COD/m^3^ day)	Total Nitrogen (mg/L)	NH_3_-N (mg/L)	Methane (CH₄) Gas Release (%)	Oil and Grease (mg/L)	°C	Working Volume (L)	HRT (day)		
1.	*Chlorella vulgaris*	95–99.9	97–99.9	ND	ND	78–98	ND	ND	ND	ND	ND	14	POME treatment by immobilization of *C. vulgaris* in alginate beads and biodiesel production.	[133]
2.	*Streptomyces platensis*	93.57 (182)	97.18 (42)	86.98 (25)	ND	ND	ND	ND	ND	ND	0.1	6	POME treatment	[135]
3.	*Chlorella sorokiniana*	90	ND	ND	ND	71	ND	ND	ND	25–30	1	15	POME treatment at 80% dilution	[136]
4.	*S. platensis*	84.9	78.3	ND	ND	91.0 (57.9)	93.8 (19.8)	ND	ND	ND	500	18	Nutrient removal	[137]
5.	*S. dimorphus*	79	71.5 (148.8)	ND	ND	87.5	88.5 (37.0)	ND	ND	ND	500	18	Nutrient removal	[137]
6.	*Chlorella pyrenoidosa*	71.43	ND	ND	ND	ND	ND	ND	ND	28 ± 1	5	14	Hybrid photo bioreactor	[138]
7.	*Chlamydomonas incerta*	67.35	ND	ND	ND	ND	ND	ND	ND	ND	0.1	28	POME treatment	[139]
8.	*Scenedesmus sp.* strain UKM9	57	86.5	ND	ND	ND	100	ND	ND	30 ± 5	1.8	30	POME treatment	[140]
9.	*Chlamydomonas* sp. UKM6	29 (1450 ± 62)	ND	ND	ND	43.50 (165 ± 12)	58.58 (84.5 ± 7.2)	ND	ND	25 ± 1	1.8	10	Biomass production and nutrient removal	[141]

°C = degree Celsius; HRT = hydraulic retention time; COD = chemical oxygen demand; BOD = biochemical oxygen demand; TSS = total suspended solids; OLR = organic loading rate; L = litre; SVI = sludge volume index; NH_3_-N = ammoniacal nitrogen; ND = no data.

## Data Availability

Not applicable.

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
