# Peer review of "Recent Developments in Biological Processing Technology for Palm Oil Mill Effluent Treatment—A Review"

_biology, 2022, doi:10.3390/biology11040525_

Round 1

Reviewer 1 Report

Wastewater treatment, in general, is of significant concern around the world since it can pose a serious threat to human health and the environment if not properly handled. It is well-known that palm oil milling activities which generate highly polluted wastewater that requires effective treatment before discharge. Whereby, highly efficient treatment technologies are required for the purification of the wastewater to achieve high-quality effluent. In that regard, the current work is pertinent and may be of interest to the journal's readers.

In this study, the authors tried to review the current state of development in the biological-based treatment technologies from palm oil milling activities. However, the manuscript has to be revised based on several issues, including;

  • In general, the manuscript should be carefully checked in terms of language and writing style. For example, in the abstract, “to discusses and demonstrates…”, L168: “are fulfill…”.
  • I would suggest removing the abbreviation in the title and because you are reviewing different technologies, is better to revise the title to reflect that.
  • It is also not preferable to use abbreviations in the abstract.
  • L24-27: The authors claim that “the most effective process for POME treatment is anaerobic treatment technique 24 using microorganisms such as Bacillus cereus MF661883, and co-culture of Clostridium butyricum LS2 and Rhodopseudomonas Biological processing technologies mooted as an efficient and sustainable management practice of POME waste”. However, there is no justification for the claim. Therefore, it is significantly important that the authors highlight the removal efficiency and optimal operating conditions of the recommended treatment approaches.
  • While writing the introductory part, the authors focused mainly on the components of the current work. Therefore, this part should be revised with more attention to materials and other efforts used in the same field of work. Hence, it is best to expand the introduction to include:
  1. Review the development of the current research problem, challenge, and current solutions
  2. More attention should be paid to the other efforts in this field
  3. Advantages and disadvantages of different treatment technologies leading to the current study. Not only the biological treatment technologies but a comparison with chemical and physical treatment approaches should be highlighted.
  • L89 is exactly similar to the first sentence in the abstract, the authors should check and revise.
  • Also, the equations in the manuscript should be reformatted to include proper numbering.
  • The discussion for figure 1 should be presented before the figure.
  • In the materials and methods section, I would suggest that the authors also include a schematic diagram of a typical biological treatment process for palm oil milling wastewater.
  • A reference should be provided for the discharge limits provided in table 1.

Author Response

Thank you for the input. Please refer the attached file.

Reviewer 2 Report

The manuscript “Recent development in biological processing technology for palm oil mill effluent (POME) treatment – A review” focuses on the bio-treatment of POME, a waste from palm oil production industry that must be properly handled before discharging. Overall, the manuscript is comprehensive and well-structured, and some of the results/conclusions, e.g., promising microorganisms, are of practical value for the countries where palm oil production is an important part of their industry. However, certain modifications could be taken to further improve the quality of this work. Thus, I would recommend this work to be published after minor revision. Below are my comments that I hope the authors would consider/address.

  1. A lot of acronyms were used in the manuscript, so I would suggest the authors provide a summary of acronyms for the readers sake.
  2. Could the authors please be more specific about how the 161 papers were selected? Also, why were the treatment technologies of POME in the past 15 years (rather than past 10 or 5 years) were selected given that the focus of this paper is recent advance?
  3. By going through the paper and examining the references, I believe the 161 papers cover both the current state-of-art full-scale POME treatment technologies and the novel lab-scale treatment technologies. Hence, the technologies discussed in the manuscript have different technology readiness levels (TRLs). The TRLs have a range between 1 to 9, with 9 being the most mature technology. It is obvious that comparing a technology with a TRL value of 1 to another with a TRL value of 9 would not lead to meaningful conclusions. I was wondering if the authors could provide an estimated TRL value for each biological processing technology listed in Table 2 through Table 6. By doing so, the readers could make easier comparisons between technologies of similar TRLs and figure out which technology is more efficient at which scale.
  4. The authors suggested some technologies/microorganisms that are more efficient in treating the POME. While I understand the COD removal rate is an important metric in wastewater treatment sector, I am not sure if COD removal rate by itself is sufficient to represent the process efficiency. In other works, I’ve seen the use of COD removal per energy consumption or per cost or per GHG emissions as an indicator for the efficiency. The use of these normalized COD removal would allow the readers explore possible tradeoffs between different technologies. I suggest the authors take another look at the choice of process efficiency indicator for this work.
  5. For tables that could not be presented within a single paper, please add the table header on each paper for the readers sake.

Author Response

Thank you for the input. Please refer to the attached file.

Reviewer 3 Report

Reviewer 1:

I recommend minor amendments at this level.

General comments:

The manuscript entitled Recent development in biological processing technology for 2 palm oil mill effluent (POME) treatment – A review” was reviewed. The work carried out in the manuscript is interesting and aimed at a summary of the biological processing technologies for the 78 POME treatments. Withal, the optimum treatment outputs with low BOD, COD, and total 79 suspended solids (TSS) will also be used as guidelines for selecting the effective biological 80 processing technologies. However, the authors are suggested to undergo several corrections as per the reviewer’s comments to improve the quality of the manuscript. Better connect your research findings to previous works published in biology and in other top journals. The innovation and the importance of this work are not clearly highlighted in the abstract, introduction, and conclusions. Please work on this and prove to us why this work is valuable. Would you explicitly specify the novelty of your work? What progress against the most recent state-of-the-art similar studies was made? Please also remove ANY lumped references. Please define each of them separately to avoid inappropriate citations. It is better to do not to use the first person's pronoun. Do not use "we, us, or our" throughout the paper. It is recommended that the authors work with a science editor who is proficient in the Native English language to improve the organization and delivery of some portions of the manuscript. This will help improve the readability and help articulate better the relevance of the authors' work.  The indication of this feature should start already from the abstract. Otherwise, not too many readers will bother reading the paper after looking at the abstract. Too many abbreviations are used in the analysis and results. I recommend a nomenclature section for the abbreviations and variables used throughout the passage. The journal's author guidelines and instructions should be followed in preparing the revised version. Other main remarks that in my opinion needs attention are the following:

Detailed comments:

Title: Ok.

Abstract:

The abstract should state briefly the purpose of the research, the principal results, and major conclusions. An abstract is often presented separately from the article, so it must be able to stand alone. In the abstract, please add an indication of the achievements from your study that are relevant to the journal scope. Please be concise - maximum 1-2 lines.

Introduction:

The review of literature needs more updating with works to have a clear and concise state of the art analysis. This should more clearly show the knowledge gaps identified and link them to the paper goals. The introduction section is poorly organized. While the general introduction is acceptable, the state-of-the-art review that follows is very difficult to understand and no specific thoughts can be inferred. The major defect of this study is the debate or argument is not clearly stated. In addition; the introduction should be clearly stated research questions and targets first. Then answer several questions: Why is the topic important (or why do you study on it)? What are research questions? What has been studied? What are your contributions? Why is it to propose this particular method?

Please eliminate the use of redundant words. Eg. In this way, Recently, Respectively, therefore, currently, thus, hence, finally, to do this, first, in order, however, moreover, nowadays, today, consequently, in addition, additionally, furthermore. Please revise all similar cases, as removing these term(s) would not significantly affect the meaning of the sentence. This will keep the manuscript as CONCISE as possible. Please check ALL. Avoid beginning or ending a sentence with one or a few words, they are usually redundant. Kindly revise all.

Please use relevant recent references by OTHER authors, recent meaning from 2018 – 2022. You may see these recent relevant articles:

https://www.mdpi.com/2071-1050/14/4/1985

https://link.springer.com/article/10.1007/s11356-021-16197-z

https://link.springer.com/article/10.1007/s10098-019-01743-8

Conclusions:

The conclusion section appears to be just a detailed summary of results/observations. All conclusions must be convincing statements on what was found to be novel, impactful based on the strong support of the data/results/discussion. Please make sure your conclusions section underscores the scientific value-added of your paper, and/or the applicability of your findings/results. Highlights the novelty of your study.

References:

Please check the reference section carefully and correct the inconsistency.
